# PGPR Modulation of Secondary Metabolites in Tomato Infested with *Spodoptera litura*

**Bani Kousar [1], Asghari Bano [2,\*] and Naeem Khan [3,\*]** 

[1]  Department of Plant Sciences Faculty of Biological Sciences, Quaid-i-Azam University, Islamabad 45320, Pakistan; bani.kousar@gmail.com

[2]  Department of Biosciences, University of Wah, Wah Cantt 47040, Pakistan

[3]  Department of Agronomy, Institute of Food and Agricultural Sciences, University of Florida, Gainesville, FL 32608, USA

\*  Correspondence: banoasghari@gmail.com (A.B.); naeemkhan@ufl.edu (N.K.)

**Abstract:** The preceding climate change demonstrates overwintering of pathogens that lead to increased incidence of insects and pest attack. Integration of ecological and physiological/molecular approaches are imperative to encounter pathogen attack in order to enhance crop yield. The present study aimed to evaluate the effects of two plant growth promoting rhizobacteria (*Bacillus endophyticus* and *Pseudomonas aeruginosa*) on the plant physiology and production of the secondary metabolites in tomato plants infested with *Spodoptera litura* (Fabricius) (Lepidoptera: Noctuidae). The surface sterilized seeds of tomato were inoculated with plant growth promoting rhizobacteria (PGPR) for 3–4 h prior to sowing. Tomato leaves at 6 to 7 branching stage were infested with *S. litura* at the larval stage of $2^{nd}$ instar. Identification of secondary metabolites and phytohormones were made from tomato leaves using thin-layer chromatography (TLC) and high performance liquid chromatography (HPLC) and fourier-transform infrared spectroscopy (FTIR). Infestation with *S. litura* significantly decreased plant growth and yield. The PGPR inoculations alleviated the adverse effects of insect infestation on plant growth and fruit yield. An increased level of protein, proline and sugar contents and enhanced activity of superoxide dismutase (SOD) was noticed in infected tomato plants associated with PGPR. Moreover, p-kaempferol, rutin, caffeic acid, p-coumaric acid and flavonoid glycoside were also detected in PGPR inoculated infested plants. The FTIR spectra of the infected leaf samples pre-treated with PGPR revealed the presence of aldehyde. Additionally, significant amounts of indole-3-acetic acid (IAA), salicylic acid (SA) and abscisic acid (ABA) were detected in the leaf samples. From the present results, we conclude that PGPR can promote growth and yield of tomatoes under attack and help the host plant to combat infestation via modulation in IAA, SA, ABA and other secondary metabolites.

**Keywords:** *Spodoptera litura* (Fabricius) (Lepidoptera: Noctuidae); *Solanum lycopersicum* L.; secondary metabolites; plant insect interactions

## 1. Introduction

*Lycopersicon esculentum* Mill. (Tomato) is one of the widely used vegetables cultivated all over the world. It is the important source of vitamin C and vitamin A [1], lycopene (carotenoids), pro-vitamin A, β-carotene and flavonoids [2]. In the recent years, its yield is significantly reduced by the infestation of leaf caterpillars.

Leaf caterpillar *S. litura* (Fabricius) (Lepidoptera: Noctuidae), also known as tropical armyworm, is among the main pests of cultivated crops that can cause significant damage to tomato crop. To this date, *S. litura* has infected about 290 plant species, belonging to 99 families [3,4]. It grows throughout

the year, and mounts nearly 7 to 8 generations per year. The larvae of *S. litura* feed initially on plant leaves and latterly feed on almost every part of the plant. The larvae can cause 12 to 23% damage to tomatoes in the monsoon and 9.4 to 27.4% in winter [5]. This insect had shown strong resistance to all conventional and some new chemically synthesized insecticides [6,7]. To combat this notorious insect attack, one can develop new insect resistant cultivars. The main drawbacks of the new cultivars' development are time and expenses. Alternatively, the use of plant growth promoting rhizobacteria having biocontrol properties is a sustainable and eco-friendly approach.

Rhizosphere bacteria form a close association with the roots of plants, they nourish on the soil nutrients and root-exudates of plants; in return they protect the host against the biotic and abiotic stresses and help in host growth [8,9]. Plant growth promoting rhizobacteria (PGPR) boost plant growth directly through the production of phytohormones and indirectly as biocontrol agents [10]. PGPR employs different mechanisms to promote plant growth and control phyto-pathogens. One of the widely recognized mechanisms is the production of inhibitory allelo-chemicals, the production of antibiotics, siderophore, lytic enzymes and the induction of systemic resistance (ISR) in host plants against a broad spectrum of pathogens [11]. Induced systemic resistance (ISR) protects the plant against a broad range of diseases [12,13], triggered by a wide variety of beneficial microbes [14].

PGPR consortium of *S. marcescens, B. amyloliquefaciens, P. putida, P. fluorescens* and *B. cereus* significantly increased the number of fruit/plant [15]. The three bacterial species viz. *B. amyloliquefaciens, B. subtilis* and *B. brevis* have significantly improved the activity of defense related enzymes in tomato plants infected with bacterial canker [16]. Several bacterial species *(Pseudomonas, Azotobacter, Azospirillum, Pseudomonas + Azotobacter, Pseudomonas + Azospirillum, Azotobacter + Azospirillum and Pseudomonas + Azotobacter + Azospirillum)* played a key role in nutrient uptake by tomato plants. Also, the rhizospheric bacteria significantly improved shoot and root dry weights, enhanced and modulated production of secondary metabolites [17] and induced resistance to various diseases [18]. *Pseudomonas aeruginosa* is an aerobic, gram-negative rod-shaped bacterium of *Pseudomonadaceae* [19] that was reported to have antifungal activity against *Fusarium moniliforme* [20]. Both *Pseudomonas aeruginosa* and *Bacillus endophyticus* were catalase and oxidase positive, solubilize phosphorus and produce bacteriocin. These bacterial strains showed significant ($p < 0.05$) increase in dry matter production, plant height and root length of maize [21]. They were found positive for the production of antibiotics [22] and had a protruding impact on plant metabolism and plant defense against environmental stresses [23,24].

The present investigation was based on the hypothesis that rhizobacteria isolated from stressed habitats can induce tolerance to plants against environmental stresses in a much better way than those from normal conditions [25]. The rhizobacteria *Bacillus endophyticus* strainY5 (Accession no. JQ792035) and *Pseudomonas aeruginosa* JYR (Accession no JQ792038) were isolated from the semiarid areas of Yousaf wala Sahiwal (15% soil moisture) and arid areas of Jhang (9% soil moisture), where maize is grown as a main crop. Soil sampling was done at the tasseling stage of maize. The role of those two PGPRs used as bioinoculants was studied on growth and yield of tomato (*Solanum lycopersicum* L.) infested with *S. litura*.

## 2. Materials and Methods

### 2.1. Plant Material

The experiment was conducted in the green house of Quaid-i-Azam University, Islamabad. Seeds of *Solanum lycopersicum L.* cv. Rio Grande was obtained from the National Agricultural Research Centre (NARC) Islamabad. Prior to sowing the seeds were surface sterilized with 70% ethanol for 2–3 min, followed by shaking in 10% clorox for 2–3 min. The seeds were finally washed with autoclaved distilled water to remove the traces of treated chemicals [13].

*2.2. Preparation of Inocula and Method of Inoculation*

Fresh cultures (24 h old) of *Bacillus endophyticus* and *Pseudomonas aeruginosa* were used to inoculate Luria-Bertani (LB) broth, incubated on a rotary shaker for 48 h at 28 °C. The cultures were centrifuged at 3000 rpm for 10 min. Supernatant was discarded, and the pellet containing the bacterial cells was suspended in the autoclaved distilled water to adjust the optical density ($\lambda = 1$) at 660 nm. The inoculum prepared was found to have $10^6$ cells/mL. Sterilized seeds were soaked in the bacterial inoculum for 3 to 4 h. The seeds soaked in autoclaved distilled water for the same period were treated as a control [5].

*2.3. Growing Conditions and the Treatments*

Seeds were sown in pots containing autoclaved sand and soil mixed in 1:3 ratio [26]. Pots were kept in the greenhouse of Quaid-i-Azam University using randomized complete block design with four replicates per treatment. The growing conditions were: photoperiod 16 h, temp 22–28 °C and humidity 60–80%.

The treatments included: Tomato seeds uninoculated uninfested control (C); Tomato seeds inoculated with *Bacillus endophyticus* (T1); Tomato seeds inoculated with *Pseudomonas aeruginosa* (T2); plants infested with *S. litura* (T3); Tomato seeds inoculated with *Bacillus endophyticus* and latterly infested the leaves at 6 to 7 branching stage with *S. litura* (T4); Tomato seeds inoculated with *Pseudomonas aeruginosa* and infested the leaves at 6 to 7 branching stage with *S. litura* (T5).

The tropical armyworm was obtained from the Insectary department, National Agricultural Research Centre (NARC), Islamabad. The leaves of tomato seedlings at 6 to 7 branching stage were infested with larvae of *S. litura* at larval stage of $2^{nd}$ instar. The larvae were starved for 2 h prior to infestation.

*2.4. Height and Weight of Plants and Weight of Tomato Fruit*

At the time of harvesting, four plants were marked from each treatment to measure the average height (cm) of the plant and their fresh and dry weights were recorded. After 180 days of sowing, the red ripened fruits were harvested and their fresh weight was measured [27].

*2.5. Physiological and Biochemical Attributes of Plants*

The physiological and biochemical parameters of leaves were measured after insect infestation.

2.5.1. Leaf Protein Content

Protein content of fresh leaves of tomato plant was estimated following the method of Lowry et al. [28], using Bovine Serum Albumin (BSA) as a standard. Fresh leaves (0.1 g) were grinded in 1 mL of phosphate buffer (pH 6.8) and centrifuged for 10 min at 3000 rpm. The supernatant (0.1 mL) was poured into the test tube and a total volume of 1 mL was made with distilled water. A mixture of 50 mL of $Na_2CO_3$, NaOH and Na-K tartrate and 1mL of $CuSO_4.5H_2O$ was added. After shaking for 10 min, 0.1 mL of Folin phenol reagent was added. The absorbance of each sample was recorded at 650 nm after 30 min incubation. The concentration of protein was determined using the following formula:

$$Protein\left(\frac{mg}{g}\right) = \frac{K-value \times dilution\ factor \times absorbance}{weight\ of\ sample}$$

K value = 19.6
Dilution factor = 2
Weight of leaf sample = 100 mg

### 2.5.2. Chlorophyll and Carotenoids Content

Estimation of chlorophyll contents was made according to the method of Arnon [29]. The tomato leaves (0.05 g) were grinded in 10 mL dimethyl sulfoxide (DMSO). The tubes were incubated at 65 °C for 4 h and then the optical density of the sample was recorded at 665 nm and 645 nm. The carotenoids content was determined following the method of Lichtenthaler and Welburn [30].

$$Chloropyll\ a\left(\frac{mg}{g}\right) = 1.07(OD_{663}) - 0.09(OD_{645})$$

$$Chloropyll\ b\left(\frac{mg}{g}\right) = 1.77(OD_{645}) - 0.28(OD_{663})$$

$$Carotenoids\left(\frac{mg}{g}\right) = Absorbance\ (OD_{663}) \times 4$$

### 2.5.3. Proline Content of Leaves (µg/g)

Free proline content in tomato plant leaves was estimated following the method of Bates et al. [31]. Fresh plant leaf (0.5 g) was grounded in 3% sulfosalicylic acids and kept overnight at 4 °C. The extract was centrifuged at 3,000 rpm for 10 min. The supernatant was mixed with acidic ninhydrin and boiled for 1 h. The solution was then cooled and toluene was added. The absorbance of the toluene layer was recorded at 520 nm against toluene blank. The content of free proline was estimated on fresh weight basis following the formula:

$$Proline\left(\frac{\mu g}{g}\right) = \frac{K - value \times dilution\ factor \times absorbance}{leaf\ weight}$$

Value of K= 17.52
Dilution factor= 2
Weight of leaf sample= 100 mg

### 2.5.4. Sugar Estimation

The colorimetric determination of total sugar (simple sugar, oligosaccharides and reducing sugar) was done following the method of Dubois et al. [32]. Fresh tomato leaves (500 mg) were grinded with 10 mL of distilled water in autoclaved mortar and pestle, centrifuged at 3000 rpm for 5 min. To the supernatant (100 µL), 1 mL of 80% (w/v) phenol and 5 mL concentrated sulfuric acid was added. The mixture was heated in a water bath till boiling and then incubated for 4 h at room temperature. The absorbance of each sample was finally measured at 420 nm.

$$Sugar\left(\frac{mg}{g}\right) = \frac{K - value \times dilution\ factor \times absorbance}{leaf\ weight}$$

Value of K = 20
Dilution factor = 10
Weight of leaf sample = 500mg

### 2.5.5. Superoxide Dismutase (SOD) Assay

The SOD activity was estimated following the method of Beauchamp and Fridovich [33].
The activity of Superoxide dismutase was expressed as units/100 g F.W.
Superoxide dismutase was calculated by the following formula:

R4 = R$_3$-R$_2$

SOD activity = $R_4/A$

$R_1$ = O.D of Reference, $R_2$ = O.D of Blank, $R_3$ = O.D of Sample

$A = R_1 (50/100)$

### 2.5.6. Determination of Indole acetic acid (IAA), Gibberellic acid (GA) and Abscisic acid (ABA) Contents

The extraction and purification for above mentioned phytohormones were made following the method of Kettner and Doerffling [34]. Plant leaves (1g) were grinded in 80% methanol at 4 °C with butylated hydroxytoluene (BHT) used as antioxidant. The extract was centrifuged and the supernatant was reduced by using a rotary thin film evaporator (RFE). The aqueous phase was partitioned 4 times at pH 2.5–3 with $\frac{1}{2}$ volume of ethyl acetate. The ethyl acetate was evaporated by a rotary thin film evaporator. The residue was re-dissolved in 1 mL of methanol (100%) and examined on HPLC (LC-8A Shimadzu, C-R4A Chromatopac; SCL-6B system controller) using UV detector and C-18 column (39 × 300 mm). The wavelength used for the detection of IAA was 280 nm and for GA was 254 nm. For ABA, the samples were injected onto a $C_{18}$ column and eluted at 254 nm with a linear gradient of methanol (30–70%), containing 0.01% acetic acid, at a flow rate of 0.8 mL min$^{-1}$ [35].

### 2.5.7. Determination of Salicylic Acid (SA) Content of Leaves

Enyedi et al. [36] and Seskar et al. [37] method was employed for salicylic acid detection. After crushing the fresh leaves (1 g) of tomato in 10 mL of 80% methanol at 4 °C. The sample was kept for 3 days with subsequent change in methanol after 24 h. The methanol was then evaporated using RFE and the residue was dissolved again in methanol, filtered and subjected to high-performance liquid chromatography (HPLC) (Agilent Technologies USA) equipped with S-1121 dual piston solvent delivery system and S-3210 UV/VIS diode array detector. Detection of SA was done at 280 nm by co-chromatography with 2-hydroxybenzoic acid as standard. The peak areas were recorded and calculated with SRI peak simple chromatography data acquisition and integration software (SRI instruments, Torrance, CA, USA).

### 2.5.8. Measurement of Shoot and Root Fresh and Dry Weights and Root Area

Shoots of 4 plants per treatment were cut at the base and weighed immediately by using the electronic balance, to measure the fresh weight of shoot. The chopped shoot was then dried at 70 °C for 72 h and dry weight was measured. The roots of the same plants were washed thoroughly with running tap water to remove soil debris. The water was absorbed on filter paper and weighed to measure the fresh weight of the root. The same root samples were used for determination of root dry weight after drying in the oven till constant weight was obtained [13]. The root area was calculated by using root law Software, Washington State University [38].

### 2.6. Thin Layer Chromatography of Methanolic Extract of Tomato Leaves

Leaves of tomato plant were harvested 24 h after infestation (80 DAS); shade dried at room temperature and grinded to fine powder. Powdered leaves (20 g) was extracted in 400 mL methanol for 72 h.The methanolic extract was dried using rotary evaporator (RFE), the residue (3 mg) was dissolved in 500 μL methanol and collected in eppendorf tube and stored at −4 °C.

Extract was spotted on a TLC plate (20 × 20 cm) coated with silica gel HF (250-350 nm). The mobile phase used was chloroform: methanol (95:5 *v/v*). The bands, representing various compounds were visualized under UV (254 nm and 380 nm) [39]. The Rf value of each band was calculated and identification of the compound from each band at specific Rf was made from the literature documented.

## 2.7. FTIR Spectroscopy

All spectra were obtained with the help of an OMNI-sampler attenuated total reflectance (ATR) accessory on a Nicolet FTIR spectrophotometer followed by the method of Lu et al. [40] and Liu et al. [41] with some modifications. Small amount of TLC eluent corresponding to the Rf value of major bands were placed directly on the germanium piece of the infrared spectrometer with constant pressure applied and data of infrared absorbance, collected over the wave number ranged from 4000 cm$^{-1}$ to 675 cm$^{-1}$ and computerized for analyses by using the Omnic software [42].

## 2.8. Statistical Analysis of Data

The data was subjected to analysis of variance using Statistix 8.1 software. The differences among various treatment means were compared using the least significant differences test (LSD) at $p \leq 0.05$ probability level (Table S1).

## 3. Results

### 3.1. Plant Growth Attributes

The plant spread, which is a measurement of plant width, was significantly (31%) higher in PGPR treated plants under unstressed condition over control (Figure 1). Insect infestation decreased the plant spread by 41%, the decrease was ameliorated by the PGPRs and the value was even greater than the control. The plant height was significantly increased in PGPR inoculated plants (Figure 1). The insect infestation significantly reduced ($p \leq 0.05$) the height of the plant by 40%, and root area by 50% of the control (Figure 1). The PGPR inoculated plants alleviated the inhibitory effects of insect infestation on plant height and root area such that the root area was significantly higher than the control. Both the shoot and root fresh weights were significantly (44% and 34%) increased in PGPR inoculated plants (Figures 2 and 3). Infestation with the insect decreased the fresh weights of both the root and shoot, the shoot fresh weight was more adversely affected. The PGPR inoculation had ameliorated the insect-induced decrease in the root and shoot fresh weight.

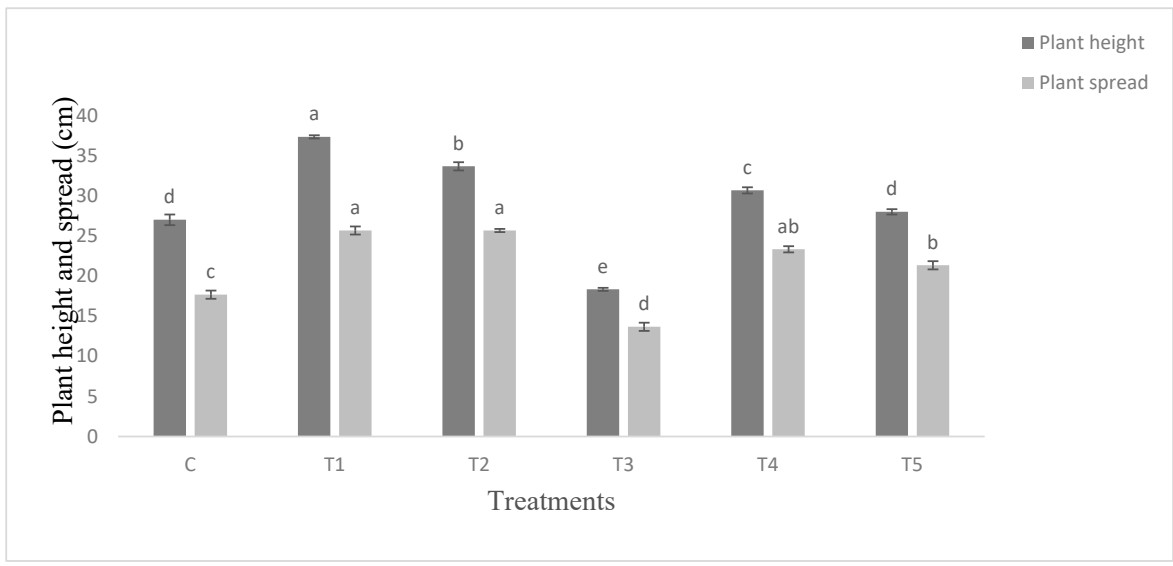

**Figure 1.** Mean plant height and plant spread (cm) of tomato under control and infested conditions. Data are means of four replicates along with standard error bars. Different letters on the bar represent significant differences ($p < 0.05$) among treatments.

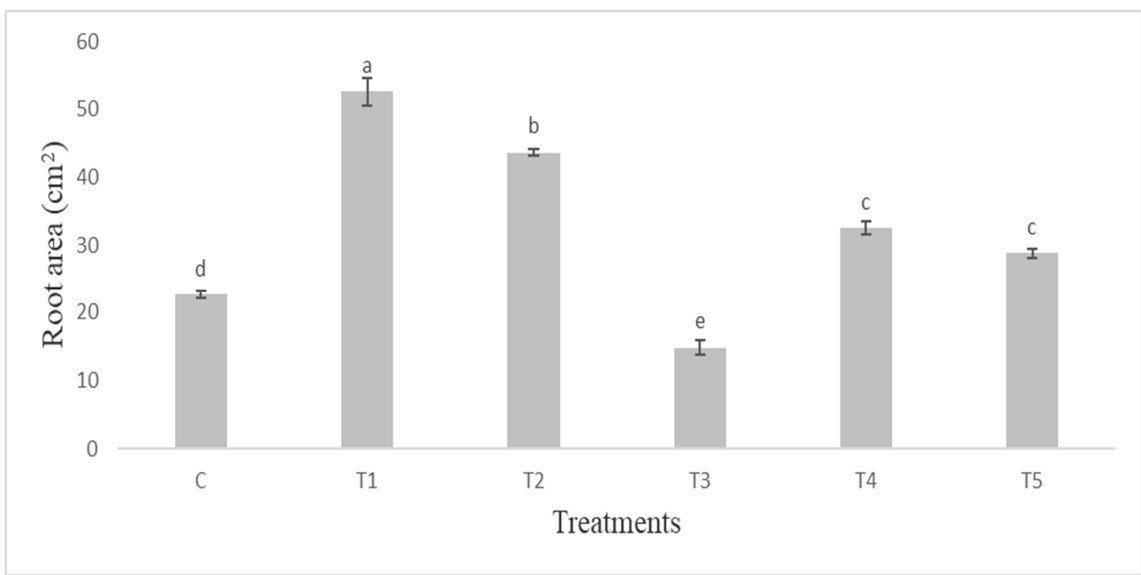

**Figure 2.** Root area (cm$^2$) of tomato plant infested with *S. litura* and under control condition. Data are means of four replicates along with standard error bars. Different letters are indicating significant differences ($p < 0.05$) among treatments.

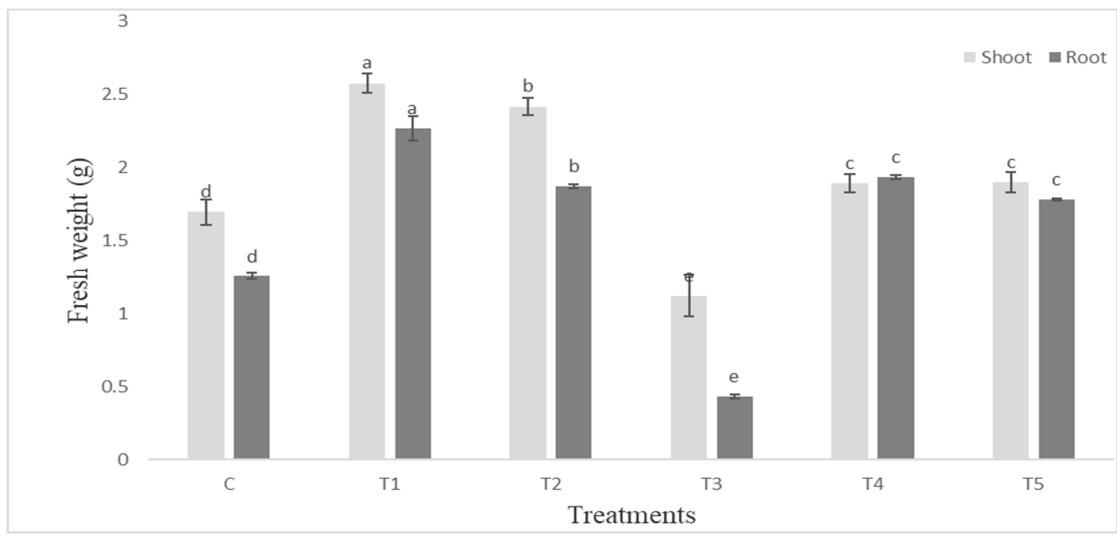

**Figure 3.** Fresh weight of shoot and root (g) of tomato plant infested with *S. litura* and under control condition. Data are means of four replicates along with standard error bars. Different letters are indicating significant differences ($p < 0.05$) among treatments.

C-uninoculated uninfested control, T1-Seeds inoculated with *Bacillus endophyticus*, T2-Seeds inoculated with *Pseudomonas aeruginosa*, T3-Plants infested with *S. litura*, T4-Seeds inoculated with *Bacillus endophyticus* and plants infested with *S. litura*, T5-Seeds inoculated with *Pseudomonas aeruginosa* and plants infested with *S. litura*.

C-uninoculated uninfested control, T1-Seeds inoculated with *Bacillus endophyticus*, T2-Seeds inoculated with *Pseudomonas aeruginosa*, T3-Plants infested with *S. litura*, T4-Seeds inoculated with *Bacillus endophyticus* and plants infested with *S. litura*, T5-Seeds inoculated with *Pseudomonas aeruginosa* and plants infested with *S. litura*.

C-uninoculated uninfested control, T1-Seeds inoculated with *Bacillus endophyticus*, T2-Seeds inoculated with *Pseudomonas aeruginosa*, T3-Plants infested with *S. litura*, T4-Seeds inoculated with

*Bacillus endophyticus* and plants infested with *S. litura*, T5-Seeds inoculated with *Pseudomonas aeruginosa* and plants infested with *S. litura*.

The dry weight of root and shoot was also higher ($p \leq 0.05$) in PGPR inoculated plants (Figure 4). The root was more responsive and the % increase in root weight was greater. The leaves were almost eaten by the insect and the shoot weight was significantly decreased to 81% whereas root weight was decreased by 38% over the control.

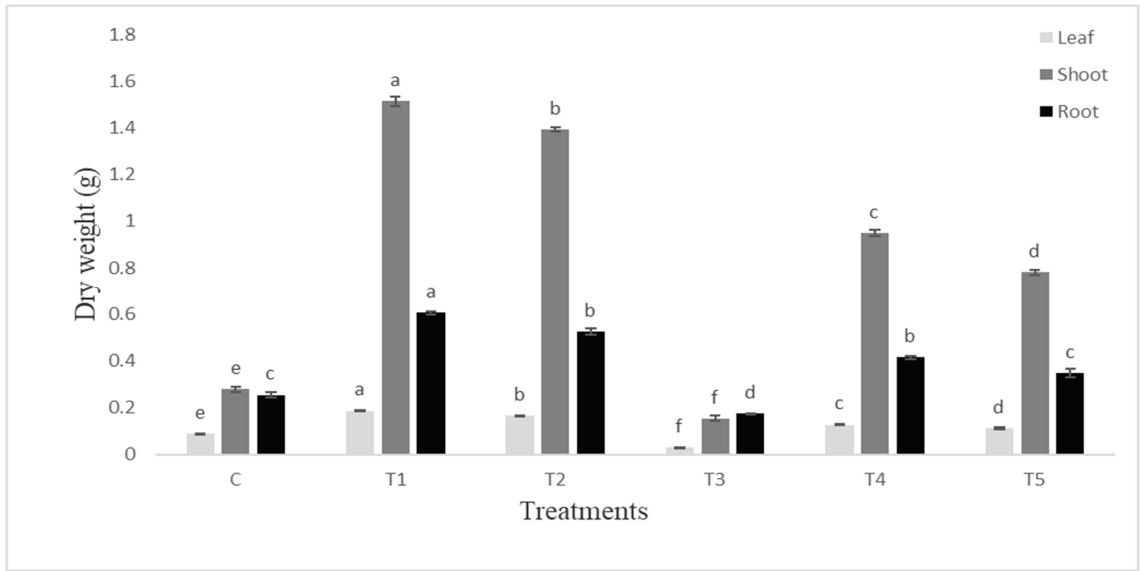

**Figure 4.** Dry weight of leaf, shoot and root (g) of tomato plant infested with *S. litura* and under control condition. Data are means of four replicates along with standard error bars. Different letters are indicating significant differences ($p < 0.05$) among treatments.

C-uninoculated uninfested control, T1-Seeds inoculated with *Bacillus endophyticus*, T2-Seeds inoculated with *Pseudomonas aeruginosa*, T3-Plants infested with *S. litura*, T4-Seeds inoculated with *Bacillus endophyticus* and plants infested with *S. litura*, T5-Seeds inoculated with *Pseudomonas aeruginosa* and plants infested with *S. litura*.

### 3.2. Physiological Parameters

The proline production was lower ($p \leq 0.05$) in the untreated control plants (Figure 5). Under unstressed conditions the PGPR treatments stimulated proline content of leaves by 18% over control. Similar percent of increase was recorded in plants infested with *S. litura*. Both the PGPR inoculated plants infested with *S. litura* exhibited marked increase in proline content of leaves over infested plants. The maximum (59%) increase was recorded in the *Bacillus endophyticus* inoculated plants infested with *S. litura*. Chlorophyll a, b and carotenoids followed the similar pattern of response to PGPR and *S. litura* infestation (Figure 6). The response of PGPR was higher ($p \leq 0.05$) particularly for carotenoids content. Both the protein and the sugar contents were higher ($p \leq 0.05$) in PGPR inoculated plants (Figure 7) under unstressed conditions. *Pseudomonas aeruginosa* showed maximum (1.4 fold) increase in sugar content over infested plants. The infestation with *S. litura* had increased sugar and protein contents significantly higher than the control. The inoculated plants receiving insect infestation exhibited up to 2.25 fold increase in sugar content as compared to that of infested plants.

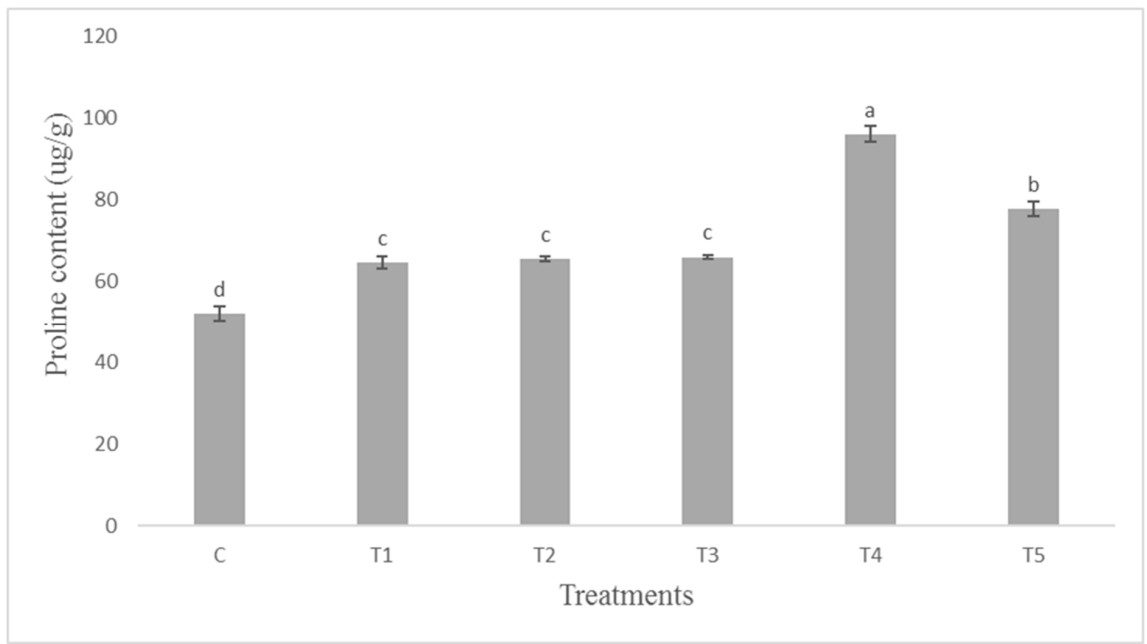

**Figure 5.** Proline content (μg/g) of tomato leaves infested with *S. litura* and under control condition. Data are means of four replicates along with standard error bars. Different letters are indicating significant differences ($p < 0.05$) among treatments.

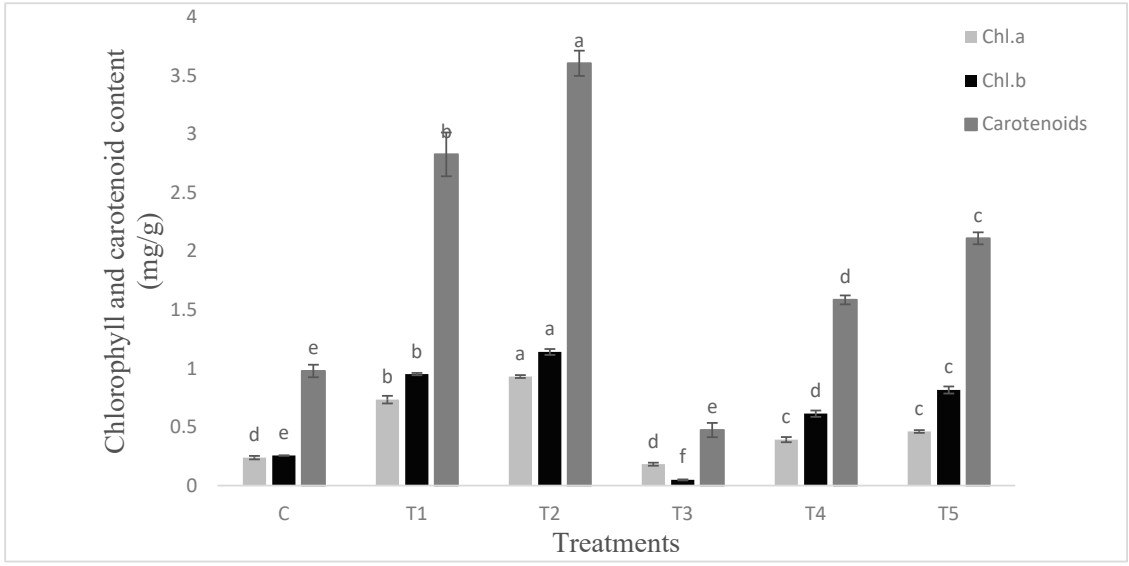

**Figure 6.** Chlorophylls and carotenoids content (mg/g) of tomato leaves infested with *S. litura* and under control condition. Data are means of four replicates along with standard error bars. Different letters are indicating significant differences ($p < 0.05$) among treatments.

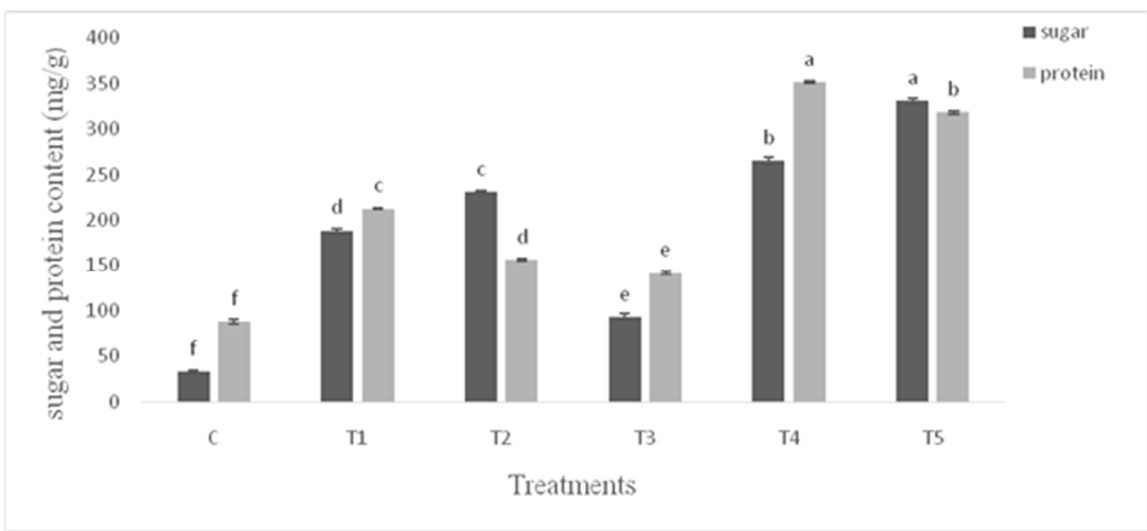

**Figure 7.** Sugar and protein content (mg/g) of tomato leaves infested with *S. litura* and under control condition. Data are means of four replicates along with standard error bars. Different letters are indicating significant differences ($p < 0.05$) among treatments.

The weight of tomato fruit was about 35% greater in plants inoculated with *Bacillus endophyticus* while *Pseudomonas aeruginosa* inoculated plants exhibited 44% increase over control. There was 26% decrease in the weight of tomato fruit in infested plants (Figure 8). The PGPR inoculated plants ameliorated the inhibitory effect of the insect and showed up to 78% increase in the fruit weight over infested plants.

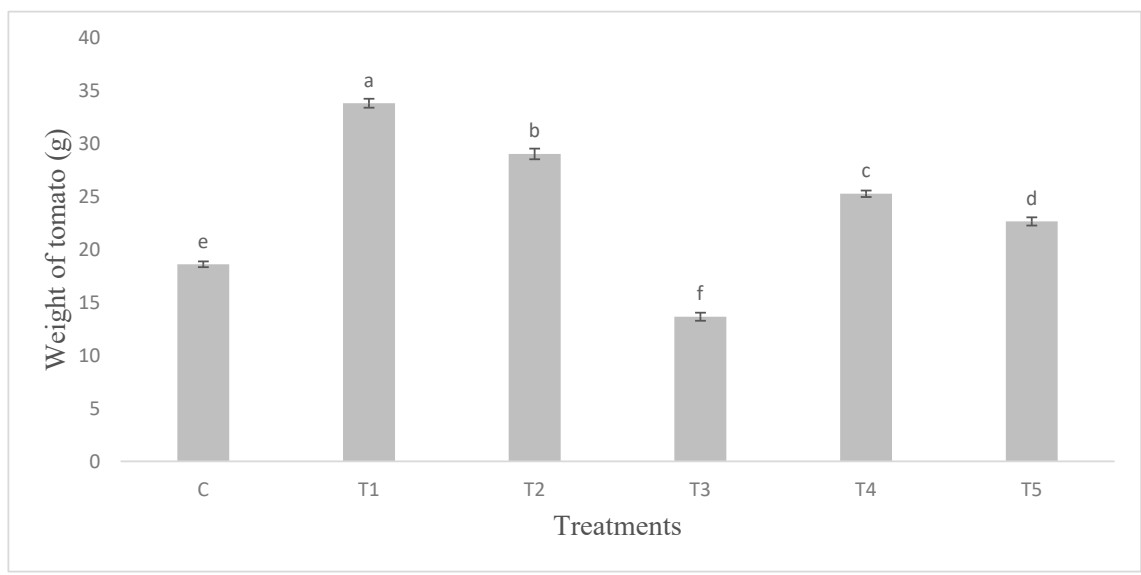

**Figure 8.** Weight of tomato fruits/plant (g) infested with *S. litura* and under control condition. Data are means of four replicates along with standard error bars. Different letters are indicating significant differences ($p < 0.05$) among treatments.

The infestation with insects enhanced the SOD activity. The SOD activity was three fold higher in leaves of plants inoculated with *Bacillus endophyticus* (T1). Plants inoculated with *Pseudomonas aeruginosa* (T2) on infestation further augmented SOD (Figure 9).

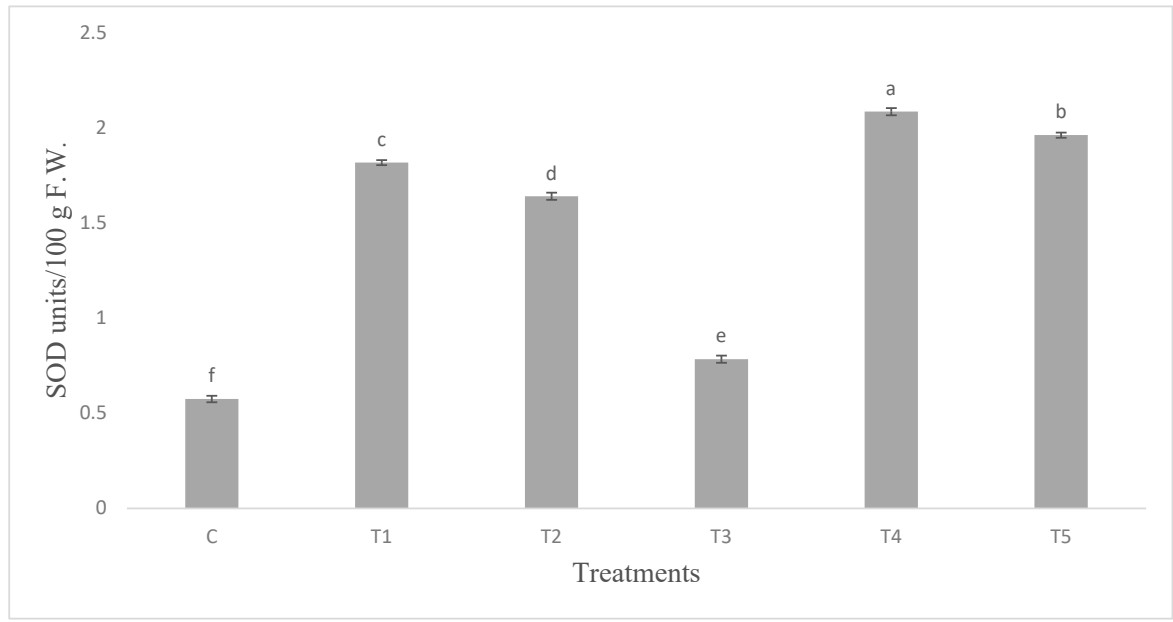

**Figure 9.** Superoxide dismutase (SOD) activity in tomato leaves infested with *S. litura* and under control condition. Data are means of four replicates along with standard error bars. Different letters are indicating significant differences (*p* < 0.05) among treatments.

### 3.3. Phytohormones Contents of Leaves

The data in Figure 10 revealed that uninoculated uninfested control leaves of tomato had traceable amounts of Salicylic acid. Insect infestation produced very little amounts of SA. Both the PGPR produced significantly higher amounts of SA in plants, *Pseudomonas* sp. being more efficient. The SA was 1.8 folds greater than infested plant leaves. In *Pseudomonas* inoculated plants, this was further augmented and significantly higher (3.6 fold) SA was recorded in infested plant leaves pretreated with *Pseudomonas aeruginosa*. IAA was not detected in the control and insect infested plants but both the PGPR produced significant amount of IAA in the leaves of inoculated plant which was further augmented and up to 449 µg IAA/g leaves was detected in the leaves of plants infested with *S. litura* and pretreated with *Pseudomonas aeruginosa* (Figure 10). Insect infestation increased the GA content of leaves significantly over control. Several fold increases in GA production were recorded in both the PGPR inoculated plants: *Pseudomonas aeruginosa* being most efficient. Both the PGPR inoculated plants overcame the insect infestation induced decrease in GA content (Figure 10). The ABA content was significantly lower in the infested plant leaves as compared to control. *Bacillus endophyticus* inoculation showed significantly higher ABA production under controlled conditions and the value was several times greater than control in the inoculated plant infested with *S. litura*.



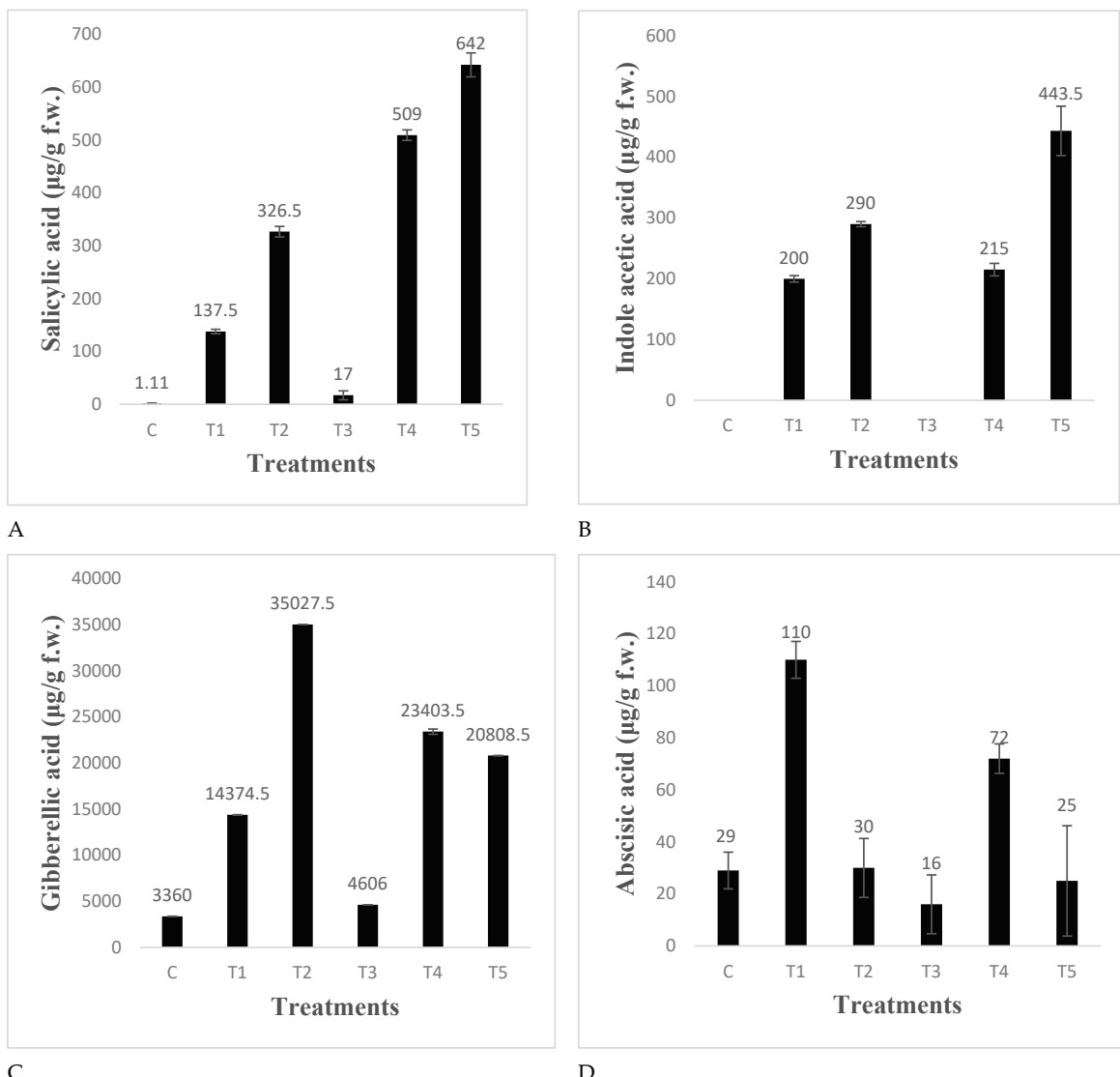

**Figure 10.** Phytohormone content in the leaves of tomato plants infested with *S. litura* and under control condition. (**A**): Salicylic acid; (**B**): Indole acetic acid; (**C**): Gibberellic acid; (**D**): Abscisic acid.

*3.4. Detection of Secondary Metabolites from Extract of Tomato Leaves*

Thin layer chromatography of tomato leaf extract showed 29 bands of different colors under UV light (Table 1). Calculated Rf values of leaf extract were compared with Rf values of standard compounds ferulic acid (0.72), salicylic acid (0.60), o-coumeric acid (0.74), trans-cinnamic acid (0.74), caffeic acid (0.85), p-coumaric acid (0.77).

The un-inoculated non infested control plant leaves extract contained caffeic acid (Rf 0.85) and quercetin (Rf 0.88). This was in contrast to *Bacillus endophyticus* inoculated plant leaves which exhibited some unidentified compounds at Rf 0.50 in addition to myricitin (Rf 0.73) o-coumaric (Rf 0.74) whereas, *Pseudomonas aeruginosa* inoculated plants showed the presence of flavonoids, ferulic acid, o-coumaric, kaempferol-7-neoheps-eridiside-glycosides in addition to some unidentified compounds of low polarity. Infestation with *S. litura* resulted in the production of caffeic acid and o-coumaric acid in addition to low and high polarity unidentified compounds. PGPR inoculated plants on infestation produced salicylic acid, rutin and kaempferol in addition to p-Coumaric acid and some unidentified compounds.

**Table 1.** Putative secondary metabolites identified on the basis of the Rf values in the extract of tomato leaves of different treatments.

| Treatments | Rf Values | Color | Compounds |
|---|---|---|---|
| Control | 0.85 | Red | Caffeic acid |
| | 0.85 | Red | Quercitin |
| T1 | 0.50 | Red | Unidentified |
| | 0.73 | Red | Myricitin |
| | 0.79 | Red | o-coumaric acid |
| T2 | 0.21 | Red | Flavonoid-glycoside |
| | 0.39 | Red | Unidentified |
| | 0.50 | Red | Unidentified |
| | 0.71 | Red | Ferulic acid |
| | 0.79 | Red | o-coumaric acid |
| T3 | 0.55 | Yellow | Kampferol-7-neoheps-eridiside |
| | 0.14 | Red | Unidentified |
| | 0.41 | Red | Unidentified |
| | 0.84 | Red | Caffeic acid |
| | 0.76 | Yellow | p-Cumaric acid |
| T4 | 0.16 | Red | Unidentified |
| | 0.23 | Red | Unidentified |
| | 0.43 | Red | Rutin |
| | 0.60 | Red | Salicylic acid |
| | 0.82 | Red | Kaempferol |
| | 0.76 | Yellow | p-Cumaric acid |
| T5 | 0.16 | Red | Unidentified |
| | 0.23 | Red | Unidentified |
| | 0.43 | Red | Rutin |
| | 0.60 | Red | Salicylic acid |
| | 0.82 | Red | Kaempferol |
| | 0.77 | Yellow | p-Cumaric acid |

*3.5. Fourier Transform Infrared Spectrometry (FTIR) of Tomato Leaves*

The data presented in Table 2 revealed that control plant (uninfested and uninoculated) leaves extract had shown the presence of amines and amides with N-H stretch and bend, aliphatic amines stretching with C-N, alkenes with C-H bend and alkyl halides with C-Cl stretch. Plants inoculated with *Pseudomonas aeruginosa* exhibited an additional bonding indicating the presence of alkynes (at frequency of 638.51 with –C≡C–H: C–H bend) which were absent in uninoculated un-infested plants. While extract of plant leaves infested with *S. litura* exhibited alkanes with C–H stretch. In addition to amines and amides with N-H stretch, aliphatic amines stretching with C-N, alkenes with C-H bend and alkyl halides with C-Cl stretching. This was in contrast to plant leaves extract previously inoculated with *Bacillus endophyticus* (T4) or *Pseudomonas aeruginosa* and infested with *S. litura* (T5) which exhibited higher frequency of N-H stretch and =C-H bend and additional bonding indicating the presence of aldehyde and amine with H–C=O: C–H stretch, N–H bend (at frequencies of 2827.91 and 1630.78) as compared to plant extract infested with *S. litura* (T5).

**Table 2.** Fourier-transform infrared spectroscopy (FTIR) of thin-layer chromatography (TLC) eluent of tomato leaves under different treatments.

| Treat. | Frequency | Bond | Functional Group | Characteristics of Peak |
|---|---|---|---|---|
| C | 3408.9 | N–H- stretch | 1°, 2° amines/amides | Medium |
| | 1632.5 | N–H- bend | 1° amines | Medium |
| | 1068.3 | C–N- stretch | aliphatic amines | Medium |
| | 967.6 | =C–H- bend | Alkenes | Strong |
| | 799.5 | C–Cl stretch | alkyl halides | Medium |
| T1 | 3412.90 | N–H stretch | 1°, 2° amines, amides | Medium |
| | 1633.25 | N–H- bend | 1° amines | Medium |
| | 1066.14 | C–N- stretch | aliphatic amines | Strong |
| | 967.48 | =C–H- bend | Alkenes | Medium |
| | 799.21 | C–Cl stretch | alkyl halides | Broad, strong |
| T2 | 3410.75 | N–H- stretch | 1°, 2° amines, amides | Medium |
| | 1633.24 | C–N- stretch | aliphatic- amines | Strong |
| | 1068.66 | C–N- stretch | aliphatic amines | Strong |
| | 967.13 | =C–H- bend | Alkenes | Medium |
| | 799.17 | C–Cl- stretch | alkyl halides | Broad, strong |
| | 638.51 | –C≡C–H:C–H- bend | Alkynes | Broad, strong |
| T3 | 3376.84 | N–H stretch | 1°, 2° amines, amides | Medium |
| | 2922.79 | C–H– stretch | Alkanes | Medium |
| | 1071.40 | C–N- stretch | aliphatic amines | Strong |
| | 966.62 | =C–H- bend | Alkenes | Medium |
| | 799.60 | C–Cl- stretch | alkyl halides | Broad, strong |
| T4 | 3412.33 | N–H- stretch | 1°, 2° amines, amides | Medium |
| | 2827.91 | H–C=O: C–H- stretch | Aldehydes | Medium |
| | 1630.78 | N–H- bend | 1° amines | Medium |
| | 1066.63 | C–N- stretch | aliphatic amines | Strong |
| | 967.01 | =C–H- bend | Alkenes | Medium |
| | 799.50 | C–Cl- stretch | alkyl halides | Broad, strong |
| T5 | 3405.01 | N–H- stretch | 1°, 2° amines, amides | Medium |
| | 1632.25 | N–H- bend | 1° amines | Medium |
| | 1066.10 | C–N- stretch | aliphatic amines | Strong |
| | 967.23 | =C–H- bend | Alkenes | Medium |
| | 799.56 | C–Cl- stretch | alkyl halides | Broad, strong |

Values are mean of 4 replications per treatment. Small amount of TLC eluent corresponding to the Rf-value of key bands were placed directly on the germanium piece of the infrared spectro-meter with persistent pressure and the infrared absorbance was collected over the wave number ranged from 4000 cm$^{-1}$ –675 cm$^{-1}$ and computerized for analyses by using the Omnic software.

## 4. Discussion

This paper evaluates the effect of PGPR as a growth promoter as well as a biocontrol agent. Although the study deals with tomatoes only in one season and also limited by the lack of behavioral study of the insect.

The growth parameters of tomato were considerably amplified after PGPR inoculation, under uninfested condition; the *Pseudomonas aeruginosa* being more effective. Of note, the effectivity of PGPR were higher under infested conditions and produce higher proline as osmoregulant, more defense hormone e.g., SA and higher level of growth promoting hormone, e.g., IAA contents and for inducing antioxidant enzyme, SOD. The PGPR inoculation not only overcame the infestation induced decrease in root and shoot weight but also increased the root and shoot weight. The PGPR effect was more pronounced on shoot dry weight. Similar results were reported by Avis et al. [43] and Babalola [44]. Shannag and Abadneh [45] reported that fresh and dry weight of shoot and root were decreased by *Aphis fabae Scopoli* in Faba Bean as compared to its respective control. Yadav et al. [46]

reported a marked increase in shoot and root dry weight in chickpea treated with PGPR. The increased weight of tomato fruit was correlated with the number of flowers, branches, plant biomass concomitant with the osmotic balance and alleviating oxidative stress. Both the PGPR were effective and significantly enhanced (≥35%) the fruit fresh weight. This increase in fruit fresh weight by the PGPR may be attributed to the fact that PGPR significantly improves the root growth and plant vigor which lead to enhanced fruit production. Fabro et al. [47] reported that tomato plants treated with PGPR showed the highest number of branches when compared to infested control. PGPR has positive effects on tomato fruit quality attributes, particularly on size and texture [48]. Widnyana [49] reported that inoculation of tomato plants with *Pseudomonas* and *Bacillus* sp. speeded up the plant growth and yield and protection against plant pathogens. Similar results that PGPR inoculation enhances the plant growth, yield and fruit weight were also reported by Almaghrabi et al. [50] and Murphy et al. [51].

Results revealed that PGPR alleviated the osmotic imbalance by increasing, proline content in the insect infested plants. The different PGPR behaved differently, both for proline production and antioxidant enzymes. It is demonstrated that *Pseudomonas aeruginosa* combats osmotic stress in infested plants through increase in sugar content as osmoregulant whereas, *Bacillus endophyticus* enhances proline content to combat osmotic imbalance. The osmotic stress is one of the secondary stresses caused by insect infestation. Proline is revealed to be an osmoregulant that accumulates in plants under a wide range of stress conditions [52,53]. It is well known that free proline accumulation in vascular plants demonstrated stresses including pathogen attack [54,55]. The accumulation of cellular osmolytes such as proline, sugar alcohols, glucosinolates etc. and soluble sugars and the expression of antioxidant systems help plants in sustaining cellular function, crucial for physiological stability of plants under stress. Ullah et al. [56] indicated that application of PGPR to plants displayed substantial increase in proline content as compared to untreated plants. Phenolics are produced by many plant species for protection against biotic or abiotic stress growth conditions and their accumulation correlates with antioxidant capacity of plants in a number of species [57,58].

Compatible solutes are used for osmotic adjustment under adverse environmental conditions [59,60]. The soluble carbohydrates in plants attacked by a fungal pathogen, as well as proportions of individual sugars, may be variously modified, both by plant regulatory mechanisms and by pathogen interference. There are several causes for quantitative and qualitative changes of sugars at the infection site. The level of sugars is reduced by their consumption for both energy and structural purposes, their uptake by the pathogen, while in autotrophic tissues it happens due to the inhibition of photosynthesis [61]. The results further demonstrate the PGPR induced changes in chlorophyll and carotenoids in normal and insect infested plants and the *P. aeruginosa* being most effective. Similar results were reported by Wang et al. [62] that PGPR isolates increased chlorophyll content significantly in tomatoes. Inoculation with *Pseudomonas* B-25 resulted in greater synthesis of chlorophyll than the diseased control. Botha et al. [63] showed that *Diuraphis noxia* feeding caused decrease in chlorophyll content in Tugela and decreased levels of chlorophyll a upon infestation [64,65].

It was demonstrated that the *P. aeruginosa* adjust osmotic stress following infestation by stimulation in antioxidant SOD activity. The PGPR effectively enhanced the SOD activity to scavenge the ROS and prevent oxidative stress in plant cells. The observed enhancement in PGPR induced SOD activity in infested plants is a mechanism to combat insect induced oxidative stress. Recently it has been reported by Sharma and Mathur [66] that PGPR alone and/or in association with fungi significantly enhanced the antioxidant enzyme activities in *Brassica juncea* infested with *Spodoptera litura* that lead to enhanced immune system against herbivory. Similarly, Zhao et al. [67] reported Aphid resistance in plants infested with *B. tabaci* nymphs, associated with enhanced antioxidant activities. They concluded that this resistance probably acted via interactions with SA-mediated defense responses.

PGPR promoted growth by nutrient acquisition and by producing bioactive compounds [68,69]. They also improve the nutrient uptake in plants by modulating plant hormones level, thereby increasing root proliferation [70]. However, response of the 2 PGPR differed substantially; *Bacillus endophyticus* exhibited lower IAA but higher GA than that of *Pseudomonas aeruginosa*. PGPR can control plant disease

directly, through the production of antagonistic compounds, and indirectly, through the elicitation of a plant defense response [71]. Fernandez-Aunion et al. [72] also reported that PGPR enhances plant growth by synthesis of bioactive compounds and activating plant defense system.

While the PGPR-elicited ISR has been studied extensively in the model plant *Arabidopsis*, it is not well characterized in crop plants. The induction of ISR was investigated by *Bacillus cereus* strain BS107 against *Xanthomonas axonopodis* pv. *vesicatoria* in pepper leaves. Choudhary and Jobri [73] demonstrated the induction of ISR elicited by *Bacillus* spp. against several fungal bacterial and viral pathogens including root knot nematodes. Yang et al. [74] reported genetic evidence of the priming effect of a *rhizobacterium* on the expression of defense genes involved in ISR in pepper. A stronger negative effect of the PGPR on the performance of leaf folder larvae was noted in rice and found that combined treatment of PGPR is more effective than individually. Several plant secondary compounds such as glucosinolates and cyanogenic glycosides yield toxic products after hydrolysis by enzymes stored and liberated during attack by chewing insects [75,76].

It is demonstrated from the present findings that SA and ABA are both involved in inducing tolerance to plants, but the mechanism of inducing tolerance against the insect varied among the PGPR used. For example, *Bacillus endophyticus* inoculation ameliorated the adverse effects of insect infestation by significantly increasing SA and ABA many folds higher than infested plants whereas, *Pseudomonas aeruginosa* ameliorated the infestation by increasing SA higher than the former strain. Salicylic acid is the integral part of signal transduction pathways initiating resistance to disease and infection [77–79]. Plant defense in response to microbial attack is controlled by signaling molecules including SA, JA and ethylene [80]. SA is an important director of pathogen stimulated systemic acquired resistance (SAR), whereas JA and ET are compulsory for rhizobacteria-mediated induced systemic resistance (ISR) [81]. Branch et al. [82] found that SA is a vital constituent of motioning the induced resistance to root-knot nematodes [83].

Plant phenolics comprises a wide array of secondary metabolites including flavonoids, Cinnamic acid, Kaempferol, Coumaric acid as well as salicylic acid synthesized to provide resistance to plants. Their number, type and concentration increase under insect attack [84] and appear to be stimulated following PGPR application. *Pseudomonas aeruginosa* produced both flavonoid glycoside and kaempferol in addition to coumaric acid whereas, *Bacillus endophyticus* had only myricitin and lack kaempferol and coumaric acid but on infestation both produced similar bioactive metabolites in plant. The chromatographic separation of leaf extract also revealed the presence of bands corresponding to Rf value of SA as well as phenolic compounds e.g., Kaempferol and coumaric acid thereby demonstrating the induction of ISR by PGPR in the inoculated plants infested with *S. litura*. Generally, the role of phenolic compounds in defence is related to their antibiotic, antinutritional or unpalatable properties. Besides their involvement in plant- animal or plant-microbe interactions; plant phenolics also play a key role as antioxidants and stress signaling [85–87]. Hammerschmidt [88] reported that phenolic metabolites are related to the resistance phenomenon of plants against their enemies.

The FTIR spectrum was used to identify the functional group of the active components based on the peak value in the region of infrared radiation. Production of alkynes and aldehyde in plants inoculated with *Bacillus endophyticus* and *Pseudomonas aeruginosa* on infestation with *S. litura* demonstrate the PGPR induced defense strategy against insects. Similar results reported by Panda and Khush [89] that chemical derived substances e.g., alkanes, aldehydes, ketones, waxes are involved in host-plant resistance to insects' function as a protective layer to save the plant. Whereas, Shavit [90] reported that inoculation of tomato plants with *P. fluorescens* WCS417r enhanced the performance of the phloem feeding insect *Bemicia tabaci*. Previously, the FTIR has been applied to classify the actual structure of certain plant secondary metabolites [91]. FTIR is one of the extensively used approaches to categorize the chemical ingredients and clarify the compounds structures [92]. The FTIR of the leaves extract revealed the presence of additional peaks of aldehyde in the FTIR of leaves of infested plants pretreated with PGPR. Chehab et al. [93] reported that aldehydes play a positive role in plant

defense. Plants defend themselves from pathogens attack by producing secondary metabolites and proteins [94,95].

## 5. Conclusions

The *Bacillus endophyticus* and *Pseudomonas aeruginosa* can be used to combat oxidative and osmotic stresses induced by *S. litura* infestation. Both the PGPRs combat insects induced adverse effects on plant growth and productivity through the production of phenolics, SA and ABA. *Bacillus endophyticus* was more effective in the improved defense strategy induction through the modulation of phytohormones and secondary metabolites. These PGPR are more effective under uninfested conditions and can be implicated as bioinoculant to endure the plants to cope better with insect infestation. Since there is alteration in the functional group and presence of aldehyde predominantly detected in plants treated with PGPR and infested with the insect armyworm. Further investigations using nuclear magnetic resonance (NMR) and liquid chromatography-mass spectrometry (LC-MS) are needed to unveil the secondary metabolites produced in PGPR inoculated plants versus uninoculated insect infested plants. Finally, an integrated approach of molecular mechanism of PGPR induced defense in plants against pests and parasites needs thorough investigation.

**Supplementary Materials:** The following are available online at http://www.mdpi.com/2073-4395/10/6/778/s1, Table S1: Results of the statistical analysis (ANOVA) for the performed parameters.

**Author Contributions:** Conceptualization, A.B.; Methodology, B.K. and N.K.; Software, B.K. and N.K.; Validation, B.K., A.B. and N.K.; Formal Analysis, N.K.; Investigation, B.K. and A.B.; Resources, A.B.; Data Curation, N.K.; Writing—Original Draft Preparation, B.K.; Writing—Review & Editing, A.B. and N.K; Visualization, B.K.; Supervision, A.B.; Project Administration, A.B. All authors have read and agreed to the published version of the manuscript.

**Funding:** This research received no external funding.

**Conflicts of Interest:** The authors declare no conflict of interest.

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
