# Peer review of "PGPR Modulation of Secondary Metabolites in Tomato Infested with Spodoptera litura"

_agronomy, doi:10.3390/agronomy10060778_

Round 1
Reviewer 1 Report
The manuscript does an excellent job demonstrating significant improvement with the inoculation of PGPR on infested tomato plants. The main contribution of this paper is the use of sustainable and eco-friendly approaches for the control of plant pests.
Please include this caption “C-uninoculated uninfested control, T1-Seeds inoculated with Bacillus endophyticus, T2-Seeds 242 inoculated with Pseudomonas aeruginosa, T3-Plants infested with S. litura, T4-Seeds inoculated with 243 Bacillus endophyticus and plants infested with S. litura, T5-Seeds inoculated with Pseudomonas 244 aeruginosa and plants infested with S. litura” in Figure 2-4.
I would suggest putting the figures after they are mentioned in the text.
Please fix Figure 7 axis title
Discussion. There was no mention of the limitations of the study
The discussion focuses on the mechanisms employed by PGPR which is not directly related to the aims of the study. Discussions should focus on explaining your results. At this stage, you cannot assume or claim which mechanism of action was employed by the PGPR without proper molecular assays.
I would like to see some discussion of the findings of the papers in relation to recent findings and developments in phenolic compounds productions in plants in response to pests/pathogens attack (this should be the focus of your discussion)
Your conclusion does not support your results. Please mention future directions or suggestions of the study.
References are adequate. Reference 1 and 72 is too old.
Author Response
The manuscript does an excellent job demonstrating significant improvement with the inoculation of PGPR on infested tomato plants. The main contribution of this paper is the use of sustainable and eco-friendly approaches for the control of plant pests.
Response: Response: We would like to thank the Reviewer for his/her evaluation and for the constructive comments and suggestions that have helped us improve the quality of the manuscript. We have revised the manuscript following your suggestions and comments to improve its quality. All the changes made in responses to the Reviewer’s comments were tracked in the revised file. We hope that our revised version will now meet your expectations. Please see below our responses enumerated to your comments and suggestions.
Please include this caption “C-uninoculated uninfested control, T1-Seeds inoculated with Bacillus endophyticus, T2-Seeds inoculated with Pseudomonas aeruginosa, T3-Plants infested with S. litura, T4-Seeds inoculated with 243 Bacillus endophyticus and plants infested with S. litura, T5-Seeds inoculated with Pseudomonas 244 aeruginosa and plants infested with S. litura” in Figure 2-4.
Response: We highly appreciate the Reviewer for this constructive comment. We have added these details with the legends of Fig 2-4.
I would suggest putting the figures after they are mentioned in the text.
Response: We would like to thank the Reviewer for this constructive suggestion. All the figures are arranged accordingly in the text.
Please fix Figure 7 axis title
Response: We are thankful to the reviewer for pointing out this mistake. We have corrected Fig.7.
Discussion. There was no mention of the limitations of the study
Response: We highly appreciate the Reviewer for this constructive comment. We have added these details to our discussion.
The discussion focuses on the mechanisms employed by PGPR which is not directly related to the aims of the study. Discussions should focus on explaining your results. At this stage, you cannot assume or claim which mechanism of action was employed by the PGPR without proper molecular assays.
Response: Based on these comments we have extensively revised the discussion part.
I would like to see some discussion of the findings of the papers in relation to recent findings and developments in phenolic compounds productions in plants in response to pests/pathogens attack (this should be the focus of your discussion)
Response: Following the Reviewer’s advice, we have carefully edited the Discussion part. In addition, the discussion has been comprehensively revised with addition of relevant references/literature in our revised manuscript. We hope that our revision would meet Reviewer expectation.
Your conclusion does not support your results. Please mention future directions or suggestions of the study.
Response: We highly appreciate the Reviewer for this constructive comment. We have edited our conclusion to make it better in the revised manuscript
References are adequate. Reference 1 and 72 is too old.
Response: We are thankful to the reviewer for this constructive comment. We have replaced the old references with the latest ones.
Reviewer 2 Report
The manuscript by Kousar et al., is of great interest for plant growth-promoting rhizobacterium. The experiments were elaborately planned. The results were well described. I have minor suggestions that can strengthen the manuscript.
“plant growth-promoting rhizobacterium” should be mentioned before using “PGPR”
2.1 Plant material: “disinfected” should be replaced with surface sterilized.
Please insert reference(s) here.
2.2. Preparation of inocula and method of inoculation: Insert references
2.3. Growing conditions and the treatments: Insert references
In 2.5.1-2.5.7: Describe how leaves were collected. How were leaves selected? By size, stage?
Line 221: please describe “The plant spread”.
Please describe statistical analysis used in this study.
Fix Figure 7
Author Response
The manuscript by Kousar et al., is of great interest for plant growth-promoting rhizobacterium. The experiments were elaborately planned. The results were well described. I have minor suggestions that can strengthen the manuscript.
Response: We would like to thank the Reviewer for his/her evaluation and for the constructive comments and suggestions that have helped us improve the quality of the manuscript. We have revised the manuscript following your suggestions and comments to improve its quality. All the changes made in responses to the Reviewer’s comments were tracked in the revised file. We hope that our revised version will now meet your expectations. Please see below our responses enumerated to your comments and suggestions.
“plant growth-promoting rhizobacterium” should be mentioned before using “PGPR”
Response: We highly appreciate the Reviewer for this constructive comment. We have explained this acronym in the abstract.
2.1 Plant material: “disinfected” should be replaced with surface sterilized.
Response: We would like to thank the Reviewer for this constructive suggestion. We made this change to our MS.
Please insert reference(s) here.
2.2. Preparation of inocula and method of inoculation: Insert references
2.3. Growing conditions and the treatments: Insert references
Response: We are thankful to the reviewer for pointing this out. All the missing references were added to the MS.
In 2.5.1-2.5.7: Describe how leaves were collected. How were leaves selected? By size, stage?
Response: We highly appreciate the Reviewer for this constructive comment. All fully expanded leaves of each treatment were collected based on their age/stage.
Line 221: please describe “The plant spread”.
Response: We highly appreciate the Reviewer’s constructive comment. Plant spread is a measurement of plant width in cm. We have added these details in the MS also.
Please describe statistical analysis used in this study.
Response: The statistical analysis is explained under subheading 2.8 and table S1.
Fix Figure 7
Response: We are thankful to the reviewer for pointing out this mistake. It has been corrected.